# Molecular Mechanisms of Kidney Injury and Repair

**DOI:** 10.3390/ijms23031542

**Published:** 2022-01-28

**Authors:** Sandra Rayego-Mateos, Laura Marquez-Expósito, Raquel Rodrigues-Diez, Ana B. Sanz, Roser Guiteras, Nuria Doladé, Irene Rubio-Soto, Anna Manonelles, Sergi Codina, Alberto Ortiz, Josep M. Cruzado, Marta Ruiz-Ortega, Anna Sola

**Affiliations:** 1Cellular Biology in Renal Diseases Laboratory, IIS-Fundación Jiménez Díaz, Universidad Autónoma Madrid, 28040 Madrid, Spain; srayego@fjd.es (S.R.-M.); laura.marqueze@quironsalud.es (L.M.-E.); irene.rubios@quironsalud.es (I.R.-S.); Mruizo@fjd.es (M.R.-O.); 2Departament of Systems Biology, Facultad de Medicina, Universidad de Alcalá, 28805 Alcalá de Henares, Spain; ASANZB@fjd.es (A.B.S.); amanonelles@bellvitgehospital.cat (A.M.); aortiz@fjd.es (A.O.); jmcruzado@bellvitgehospital.cat (J.M.C.); 3Department of Pharmacology, Facultad de Medicina, Research Institute Hospital La Paz (IdiPAZ), Universidad Autónoma de Madrid, 28029 Madrid, Spain; raquel.rodrigues@uam.es; 4Centro de Investigación Biomédica en Red enfermedades Cardiovasculares (CIBERCV), Av. Monforte de Lemos, 28029 Madrid, Spain; 5Division of Nephrology and Hypertension, IIS-Fundación Jiménez Díaz, Universidad Autónoma Madrid, 28040 Madrid, Spain; 6Department of Nephrology and Transplant Group, Institut d’Investigació Biomédica de Bellvitge (IDIBELL), Hospitalet de Llobregat, 08907 Barcelona, Spain; r.guiteras@gmail.com; 7Vascular and Renal Translational Research Group, Institut de Recerca Biomèdica de Lleida IRBLleida, 25198 Lleida, Spain; nuriadolade@gmail.com; 8Department of Nephrology, Hospital Universitari Bellvitge, 08907 Barcelona, Spain; scodina@bellvitgehospital.cat

**Keywords:** acute kidney injury, chronic kidney disease, regeneration, kidney, molecular mechanisms, cell death, treatment

## Abstract

Chronic kidney disease (CKD) will become the fifth global cause of death by 2040, thus emphasizing the need to better understand the molecular mechanisms of damage and regeneration in the kidney. CKD predisposes to acute kidney injury (AKI) which, in turn, promotes CKD progression. This implies that CKD or the AKI-to-CKD transition are associated with dysfunctional kidney repair mechanisms. Current therapeutic options slow CKD progression but fail to treat or accelerate recovery from AKI and are unable to promote kidney regeneration. Unraveling the cellular and molecular mechanisms involved in kidney injury and repair, including the failure of this process, may provide novel biomarkers and therapeutic tools. We now review the contribution of different molecular and cellular events to the AKI-to-CKD transition, focusing on the role of macrophages in kidney injury, the different forms of regulated cell death and necroinflammation, cellular senescence and the senescence-associated secretory phenotype (SAPS), polyploidization, and podocyte injury and activation of parietal epithelial cells. Next, we discuss key contributors to repair of kidney injury and opportunities for their therapeutic manipulation, with a focus on resident renal progenitor cells, stem cells and their reparative secretome, certain macrophage subphenotypes within the M2 phenotype and senescent cell clearance.

## 1. Introduction

Chronic kidney disease (CKD) is a growing disease burden, predicted to become the fifth global cause of death by 2040 [1]. The causes of CKD can be genetic, metabolic, immunological and vascular, among others, with metabolic (i.e., diabetes) and vascular causes being the most common [2,3]. Therefore, different mediators and molecular mechanisms participate in the progression of kidney damage (Figure 1) [2,3]. However, CKD is associated with an increased risk of CKD progression, death and acute kidney injury (AKI), which is independent from the cause of CKD. AKI may also have different causes, classically classified as prerenal (decreased perfusion), renal (parenchyma cell injury) and post-renal (urinary tract obstruction). Renal causes include untreated and persistent obstruction or decreased perfusion, as well as tubular, glomerular and interstitial injury. A suboptimal array of early diagnostic biomarkers and of effective treatments for AKI or CKD contributes to the increasing global impact of kidney disease. Therefore, there is an urgent unmet medical need to further understand the mechanisms of kidney damage and repair to develop novel biomarkers and therapies that improve the outcomes of kidney diseases.

There is a well-established link between AKI and CKD. CKD predisposes to AKI and AKI, especially repeat or severe AKI episodes, which in turn promote CKD progression [2,3]. Beyond the severity or persistence of AKI, there are no good biomarkers that predict the AKI-to-CKD transition and may guide the enrollment of early AKI patients in clinical trials aimed at preventing this transition. Thus, even potentially reversible causes of AKI, such as obstruction, may evolve into CKD if obstruction is not relieved within a certain time frame, as has been well documented experimentally [4]. Failure of serum creatinine to decrease is currently the best early indicator of the AKI-to-CKD transition, but it is still a late event that does not allow the initiating of preventive strategies when AKI is diagnosed. This raises the issue of defective kidney repair following AKI and/or a negative impact of prior CKD on kidney repair. The adult kidney displays repair mechanisms after AKI but has limited cell turnover. The precise mechanisms of kidney repair remain to be fully understood as they are related to the relative contributions of specialized renal progenitor resident cells or the division of de-differentiated mature cells. In addition, the molecular footprints of AKI and biological processes associated with unsuccessful attempted repair can become a threat to the organ.

The acute phase of AKI, characterized by cell death, is followed by a recovery phase, where there is an activation of protective and regenerative mechanisms in surviving cells to restore epithelial cell properties and functions [3,5]. However, failed resolution may result in partial epithelial–mesenchymal transition (EMT), cell senescence, G2/M cell-cycle-arrest (CCA), activation of fibroblasts and immune cells and epigenetic modifications that perpetuate inflammation and release a profibrogenic secretome, contributing to the progressive decline of kidney function and fibrosis [3,5,6] (Figure 1). In this review, we examine key molecular and cellular mechanisms involved in both maladaptive responses and the endogenous repair mechanisms in the kidney, which offer an opportunity for the design of new therapeutic approaches.

## 2. Molecular and Cellular Events Contributing to the AKI-to-CKD Transition

Key contributors to kidney injury and to the AKI-to-CKD transition include macrophages, the different forms of regulated cell death and necroinflammation, cellular senescence and SAPS, polyploidization, and podocyte injury and activation of parietal epithelial cells.

### 2.1. Macrophages in Kidney Injury

Inflammatory cell recruitment is an early event in AKI. Neutrophils are first recruited and favor the immediate recruitment of macrophages. These recruited neutrophils have a short life [7], whereas recruited macrophages and resident macrophages can persist for long periods in glomeruli or the tubulointerstitium. Macrophages have a critical role in wound healing and promote regeneration by bridging the initial inflammation and later tissue regeneration and repair phases. However, persistent or long-lasting macrophage presence in tissues may extend the damage phase, which finally leads to a failure of tubular repair, resulting in maladaptive kidney repair and contributing to the AKI-to-CKD transition [8,9,10].

Macrophages have a great plasticity to adapt to the environment, and their role in damage and/or repair is phenotype-dependent [11,12,13]. Thus, activation or deactivation of macrophages by different stimuli may determine the balance of healing versus injury. M1 macrophages have been proposed as inflammatory cells triggering kidney damage through pro-inflammatory cytokine secretion, including IL-6, TNFα and IL-1β [14,15] while M2 macrophages are involved in kidney tissue repair [16]. Different macrophage types coexist in the different disease phases of inflammation, repair/regeneration and fibrosis, but the M1/M2 ratio changes over time [12].

Pro-inflammatory macrophages have a key role in the onset of murine kidney disease through the specific transmembrane pattern recognition receptor Mincle [17], the M1 polarization regulator KLF4 [18] and/or the release of the specific macrophage inflammatory factors HMGB1, IL-1β and TNFα [19,20,21,22]. Moreover, damaged tubular epithelial cells communicate with and recruit M1 macrophages through miR-23a-enriched exosomes [23] or damage-associated molecular patterns (DAMPs) [24], promoting tubulointerstitial inflammation. Thus, M1 depletion protects against kidney ischemia-reperfusion injury (IRI) in vivo [12], while the transference of M1 macrophages increases kidney injury [24].

M2 macrophages are classically considered anti-inflammatory cells. However, M2 macrophages are also involved in kidney fibrosis in mice [13,25], and in patients with type-2 diabetes there was a positive correlation between kidney CD163+ (M2) macrophages and fibrosis [26].

Hence, the study of the role of renal resident and/or infiltrating monocytes, macrophages and dendritic cells is complicated by the overlapping phenotypic heterogeneity, phenotypic transitions, surface marker expression and functional diversity in the kidney during health and injury [27]. Independent of the exact phenotype, however, a common feature is their close association with kidney fibrosis [28].

The AKI-to-CKD progression is more likely in the elderly. This may represent the existence of subclinical pre-existent CKD and also underlying maladaptive repair mechanisms related to aging kidneys. In a recent study in aging mice, defective macrophage-M2 polarization contributed to persistent chronic inflammation after IRI, which played an important role in the progression of fibrosis and the decreasing of GFR. Interestingly, defective CSF-1 signaling mediated persistent M1-mediated inflammation and impaired macrophage M2 polarization [29].

### 2.2. Regulated Cell Death

Renal tubular cell death is a key process during AKI. Indeed, tubular necrosis and loss of brush border showed the best correlation with renal dysfunction in biopsies of AKI patients [30,31]. The classic paradigm is that after an initial wave of tubular cell death, surviving tubular cells’ dedifferentiation and self-duplication potentially overshoot, and a second wave of cell death adjusts the final cell numbers [32]. First, apoptosis was described as the main form of cell death involved in AKI [33]. However, in the last decade, there is increasing evidence for a key contribution of regulated necrosis, such as necroptosis or ferroptosis, to cell death during AKI [34,35]. Apoptosis and necrosis are interconnected, and inhibition of one could activate the other [36,37]. However, the impact on inflammation and further tissue injury clearly differs between apoptosis and necrosis. Ferroptosis and necroptosis may occur synchronously, such as in kidney IRI [38,39], or sequentially, such as in folic acid nephropathy [37,40]. Necroptosis and ferroptosis also contribute to oxalate nephropathy and necroptosis is involved in cisplatin-induced AKI [41,42,43].

Unlike cells dying from apoptosis, which display cell surface “eat-me” signals and are rapidly engulfed by macrophages or adjacent healthy cells, cells dying from regulated necrosis release DAMPs and alarmins that amplify tissue injury, in a process termed necroinflammation that interferes with kidney repair [44]. DAMPs and alarmins activate the innate immune system, triggering a stronger inflammatory response that alters the tubular microenvironment, promoting a profibrotic phenotype in tubular cells [45,46,47] and recruiting profibrotic macrophages [48,49]. Necroinflammation also induces an amplification loop of inflammation-related tubular cell death, leading to maladaptive repair [46]. In folic acid nephropathy, an initial ferroptosis wave triggers TWEAK-dependent necroinflammation followed by a necroptosis wave that leads to persistence of AKI [37,40,50]. In this regard, necroptotic proteins RIPK3 or MLKL mediate early kidney injury after IRI and amplify injury through necroinflammation, leading to kidney fibrosis in long-term IRI [51]. RIPK3 also promoted fibrosis in a 28-day folic acid nephropathy model, suggesting a role of this protein in AKI–CKD transition; however, it is not clear whether this effect of RIPK3 is dependent on the necroptosis pathway [52], as bone marrow-derived RIPK3 mediates kidney inflammation in AKI independently from necroptosis [53].

Overall, the different forms of regulated necrosis contribute to kidney injury and may potentially drive the AKI-to-CKD transition, but they may also link to kidney repair through the recruitment of macrophages and the role of apoptosis to clear excess or damaged cells, including immune cells and fibroblasts.

### 2.3. Cellular Senescence

Cellular senescence may be triggered by different stimuli, such as oncogene induction, DNA damage, telomere shortening, inflammation, reactive oxygen species production and toxins [54]. In the kidney, cellular senescence has been described in different states of kidney repair and regeneration, including early AKI, the AKI-to-CKD transition, CKD progression and transplant rejection [55,56].

There are several senescence-related mechanisms (Figure 2). One of the most studied is the DNA-Damage-Response (DDR) pathway, which is activated by cellular DNA damage, as a protective process. DDR activation provokes cell cycle arrest. Initial steps include the phosphorylation of the histone H2AX (γH2AX), which is used as a DDR marker, and activates the inhibitors of the cyclin-dependent-kinases (CDKs), p16ink4a and p21cip1, via p53 induction [55,57,58,59]. The final step in this process is the inhibition of CDK-mediated phosphorylation of the retinoblastoma tumor suppressor (Rb). Then, Rb hypo-phosphorylation prevents cell cycle progression, blocking cellular proliferation [55]. After DNA repair, cells abandon this growth arrest state and re-enter the cell cycle [60]. However, in pathological conditions, DDR and CCA are permanent, thus provoking cellular senescence [61]. Senescent cells accumulate in kidney disease and the activation of senescence mechanisms in tubular cells leads to the failure of regeneration following AKI or the AKI-to-CKD transition [29,62]. Proximal tubular cells in CCA suffer profound changes in intracellular signaling, provoking phenotypic alterations characterized by an aberrant and detrimental secretome termed the senescence-associated secretory phenotype (SASP) [59,63]. Glomerular cells, such as podocytes and endothelial cells, may undergo senescence, leading to SASP, excessive ECM accumulation (Figure 2) and glomerulosclerosis [6].

Cells of the immune system can also become senescent. The concept of immunosenescence refers to altered immune cells that play a deleterious role promoting tissue inflammation, termed inflammaging [64,65]. In murine folic acid-induced AKI, aged kidneys were predisposed to more severe injury and inflammation due to baseline senescence and inflammaging, when compared to young kidneys [56].

### 2.4. The Senescence-Associated Secretory Phenotype (SAPS)

The SASP is a key feature of senescent cells, enriched in pro-inflammatory cytokines, growth factors and other compounds. By releasing these factors, senescent cells can exert deleterious pro-inflammatory and pro-fibrotic actions [54,63,66]. The proinflammatory factors found in the SASP include IL-6, IFN-γ, IL1β, TNF-α or MCP-1 [57]. As discussed above, transient inflammation participates in kidney repair, whereas persistent inflammation contributes to kidney injury progression and kidney failure [67,68,69]. MCP-1/CCL2 levels correlate with loss of kidney function and this chemokine was proposed as a biomarker of kidney fibrosis [70]. MCP-1/CCL2 could be a biomarker of kidney fibrosis and function decline [71,72]. SASP pro-inflammatory cytokines activate the nuclear factor-kappa B (NF-κB) signaling pathway [73,74], one of the most important drivers of inflammation and loss of nephroprotective factors, such as Klotho [75]. NF-κB is also activated in senescent cells, being the main regulator of SASP [57].

In age-related diseases, including CKD, the NLRP3-inflammasome is activated and contributes to inflammaging [76]. The NLRP3-inflammasome activates caspase-1 to produce SASP components and other cytokines such as IL-1α, IL-1β and IL-18 [77,78,79]. In preclinical studies, overexpression of proinflammatory SASP genes was associated with increased senescence-related mechanisms and kidney injury following toxic or IRI AKI [56,80,81].

SASP components also include profibrotic factors, such as transforming growth factor-β1 (TGF-β1), cellular communication network factor 2 (CCN2/CTGF) and PAI-1 [66]. TGF-β1 is considered a master promoter of kidney fibrosis through the activation of canonical pathways, as well as the phosphorylation and activation of Smad2/3 proteins [82]. TGF-β1 activates fibroblasts and myofibroblasts, increases collagen production and accumulation and regulates the expression of tissue inhibitors of MMPs. The transition of bone marrow-derived macrophages into collagen-producing myofibroblasts contributes to the accumulation of ECM proteins in the kidney. An elegant study using chimeric mice and in vitro studies clearly demonstrated the role of TGF-β/Smad3 signaling in this phenotype transition [83]. CCN2 is a matricellular ECM protein that participates in diverse biological processes, including cell proliferation and migration, ECM remodeling and kidney fibrosis [3,6,84,85]. A key paradigm in fibrosis was to consider CCN2 a key downstream mediator of pro-fibrotic factors, including TGF-β1 and Angiotensin II [86,87]. CCN2 is widely used as a fibrotic marker in kidney tissue. CCN2 promotes partial EMT in tubular cells and ECM accumulation in cultured renal cells, and is considered a therapeutic target in kidney fibrosis [6]. SASP pro-fibrotic factors also upregulate ECM components in kidney parenchymal cells and promote partial EMT in tubular cells, contributing to the AKI-to-CKD transition or CKD progression, leading to kidney fibrosis [3,88].

The SASP also negatively regulates cell growth, migration, and the differentiation of adjacent, non-senescent cells in a paracrine-manner. Thus, senescent cells may spread the maladaptive phenotype to neighboring cells, a process known as secondary senescence [66] that promotes the progression of kidney injury, inflammation and fibrosis [55,89,90].

### 2.5. Polyploidization: A Question of Size; Samson or Delilah

Tissue size is controlled by two biological strategies. One is through cell number, where cell proliferation directly affects tissue size and scaling, and the other is through cell size, where the DNA contents may increase through polyploidy. During repair, polyploid cells will occupy a large tissue space that might have been freed up by loss of other cells. This strategy would compensate cell loss and promote wound repair. Therefore, polyploidy plays a role in tissue repair and organ regeneration [91,92].

Cardiomyocytes and hepatocytes may undergo both proliferation and polyploidization. Direct observation of labelled cells during regeneration after partial hepatectomy disclosed that hepatocytes initially increase in size through ploidy, while cell proliferation contributes to hepatocyte regeneration only if most of the liver is removed [93,94,95].

However, polyploidization may impair tissue repair in some organs or circumstances. Thus, following completion of heart regeneration, polyploid cells are cleared by apoptosis and replaced by dividing cells [92].

In the case of the kidney, recently, Lazzeri et al. [96] described that PTC also becomes polyploidy after some insults, but the question is whether these alternative cell cycles contribute to repair or to a secondary event of damage. On the one hand, the researchers demonstrated that, as well as in the liver, the majority of tubular epithelial cells entering the cell cycle after AKI undergo hypertrophy, which would maintain (on a limited basis) the renal function, but would not promote repair. Potentially, this cell cycle arrest of tubular cells after acute damage could provide important protection by preventing the division of potentially damaged cells [97] but, at the same time, prolonged cell cycle detection may promote a pro-fibrotic and senescent phenotype by amplifying the secondary events of damage and promoting the AKI-to-CKD link.

Polyploidization of podocytes is a well-known feature [98]. Adult podocytes characteristically cannot complete cytokinesis although they can undergo a full cell cycle including mitosis, with a consequent increase in DNA content without cell division that restores a normal podocyte number after injury [99,100]. The increase in size is deleterious, as podocytes sit outside the glomerular capillaries and are exposed to the hydrostatic pressure that drives glomerular filtration and pushes them into the urinary space. During CKD progression, loss of nephrons leads to a compensatory increase in the size of remnant glomeruli as well as to compensatory single nephron hyperfiltration, i.e., an increase in hydrostatic pressure. The combination of relative podocytopenia and increased pressures facilitates the loss of enlarged podocytes [101]. In diabetic nephropathy, the increase in glomerular size and resulting relative podocytopenia is an early phenomenon, which is also well described in Fabry nephropathy [102]. Abnormalities of podocyte mitosis, the increase in mitotic cell cycle proteins despite the lack of proliferation in vitro and also podocyte polyploidy in vivo were observed in experimental membranous nephropathy [103,104]. However, no interventional studies have yet targeted podocyte polyploidy to slow the transition from acute podocyte injury to CKD or CKD progression.

### 2.6. Podocyte Damage and Activation of Parietal Epithelial Cells

Physiological stress or pathological stimuli cause podocyte injury and modify podocytes phenotypes. This response may be maladaptive, characterized by modifications in biological processes, including cellular metabolism, dysregulation and loss of integrity [105]. Mechanical stress induced by increased trans-capillary pressure can damage the podocyte, causing cell hypertrophy and cytoskeletal dysregulation [106]. As discussed above, glomerular hyperfiltration induces shear stress in podocyte foot processes and body surfaces that are magnified for large podocytes [107].

During glomerular injury, mediators such as inflammatory cytokines (e.g., TWEAK, MIF, TGF-β1, MCP-1 and IL-1β), complement, angiotensin II and metabolites (e.g., high glucose concentrations, lyso-Gb3 and 3,4-DGE) cause podocyte injury and contribute to podocyte loss, activating intracellular pathways such as oxidative stress and Notch signaling as well as adaptive responses such as HSP27/HSPB1expression [108,109,110,111,112,113,114,115]. Indeed, podocytes are also targets, in at least certain forms, of AKI, such as pigment-induced-AKI or IRI, considered so far as mainly driven by tubular cell injury [116,117].

Oxidative stress causes podocyte injury in AKI, namely diabetic or hypertensive nephropathy, triggering Rac-1 GTPase signaling, FOXO3a activation, endoplasmic reticulum stress and mitochondrial damage, among others [118,119,120,121,122,123,124,125,126].

Glomerular PECs are involved in fibrosis, leading to crescents in rapid progressive glomerulonephritis/crescentic glomerulonephritis (RPGN/CGN) such as ANCA-associated glomerulonephritis, and to focal and segmental glomerulosclerosis (FSGS) [127,128,129,130]. During crescent formation, activated PECs invade the glomerular urinary space, irreversibly distorting the glomerular structure and obstructing the urinary space [131,132,133]. PEC “activation” implies a phenotype change that promotes their abnormal proliferation and their tuft-oriented migration towards the urinary space of the glomerulus. Activated PECs have an immature progenitor-like phenotype and a higher capacity for hyperplastic expansion (CD44+/cytokeratin-) and associate with inflammatory cells, mainly monocytes/macrophages CD68+) (Figure 3).

CD44, CD9 and the glucocorticoid receptor contribute to PEC hyperplasia and activation. CD44-deficient mice had milder glomerular cell proliferation, crescent formation and collapsing FSGS [134]. In murine CGN and FSGS, the PEC-specific gene blockade of Cd9 preserved glomerular injury through reduced expression of CD44 and β1 integrin as well as the migration of PECs into the glomerular tuft [135]. Surprisingly, given that CGN is treated in humans with glucocorticoids, glucocorticoid receptor inhibitors protected against murine CGN by decreasing cellular crescent formation and PEC proliferation/migration, uncovering the direct effects of glucocorticoids on PEC in addition to their well-known immune-suppressive actions [136]. Additionally, PECs are, together with podocytes, a source of glomerular heparin-binding epidermal growth factor-like growth factor (HB-EGF) during CGN. HB-EGF activation of the EGF receptor modifies podocyte phenotypes and favors CGN [137]. This points to a podocyte–PEC interaction; indeed, PECs are present in the neighborhood of injured podocytes [105,138,139,140], supporting the hypothesis that injured podocytes release soluble mediators that induce PEC activation and migration towards the inside of the glomerulus [141].

## 3. Recovery and Repair Strategies

Key contributors to the repair of kidney injury include resident renal progenitor cells, stem cells and their reparative secretome, certain macrophage subphenotypes within the M2 phenotype and senescent cell clearance.

### 3.1. Resident Renal Progenitor Cells

Both circulating bone marrow stem cells and renal resident stem cells from different locations are thought to contribute to kidney repair and regeneration [142,143,144]. Renal progenitor cells, such as CD133+ CD24+ cells, have the ability to self-renew and to differentiate into different kidney cell types as well as to adipogenic, osteogenic and chondrogenic lineages [145,146].

ACE inhibitors [147], retinoids [148] or corticosteroids [149] increased podocyte numbers after depletion in the absence of proliferation, suggesting another cell source for novel podocytes. A bone marrow origin of circulating progenitors for podocytes is not clearly substantiated in experimental and human disease [150]. Podocytes and PECs originate from the same pool of mesenchymal cells and PECs could return to a fetal developmental reprogramming and become podocytes [151]. Indeed, in the glomerular vascular pole, a small population of PECs (“glomerular epithelial transitional cells”) co-express both podocyte and PEC proteins. Depending on the specific proteins expressed, these possible progenitor cells will be catalogued as “ectopic podocytes” or “parietal podocytes” [129,152]. Cultured CD133+ CD24+ PECs obtained from the tubular pole of the Bowman’s capsule need retinoids for normal survival and function [153], but have a multi-progenitor potential and can be differentiated into podocytes or tubular cells [146,154,155]. These findings are in line with observations that PECs could be precursor cells of podocytes in some renal diseases such as diabetic nephropathy (DN) or FSGS [156,157,158,159,160,161,162]. In experimental FSGS, PECs began to express podocyte proteins, and acquired podocyte phenotypes such as a decreased Notch signaling [163]. The chemokine stromal-derived factor (SFD/CXCL12) blockade increased renal progenitor differentiation towards a podocyte phenotype [164]. Migration of PECs to the glomerulus to replace senescent podocytes would be beneficial, but aberrant repair by PECs may be to contribute to podocyte damage and crescent formations, as detailed above [165,166,167]. Hence, is necessary to elucidate which secreted molecules from damaged podocytes recruit PECs to repopulate the glomerulus as opposed to signals that promote crescent formation.

CD133+ CD24+ CD106+ renal progenitor cells from proximal tubules exhibit a higher rate of proliferation, self-renewal and differentiation than CD133+ CD24+ CD106- from Bowman’s capsule, despite sharing some markers such as vimentin and cytokeratin-7 [168], and also improve kidney function after kidney injury [154,168,169]. Lineage tracing identified a Sox9+ progenitor cell population in a proximal tubule involved in AKI recovery [170]. Other progenitor cells from tubules, especially from proximal tubules, express CD90, PAX-2 and CD44 but not CD133 [145,154,171], and are capable of differentiation into tubular epithelial cells and the mesodermal and chondrogenic lineages [145]. Tubular progenitor cells are more resistant to death and undergo mitosis to replace cells lost during injury [96].

### 3.2. Stem Cells and the Reparative Secretome

Kidney injury results in a local microenvironment that recruits and activates progenitor cells and cell-dependent tissue repair [144]. The regenerative potential of pluripotent embryonic (ESCs) or inducible (iPSCs) stem cells and mesenchymal stem cells (MSCs) is under study to repair experimental AKI and CKD [172,173]. MSC from different origins, including bone-marrow, the umbilical cord, amniotic fluid and adipose tissue [173,174], and renal progenitor cells (NPC) derived from human iPSCs [175,176] are renoprotective in experimental AKI and CKD. The renoprotective effects observed after stem cells’ transplantation depend on paracrine effects rather than on the integration of cells into host damage tissue [172,173]. These paracrine effects are mediated by the reparative secretome, i.e., by the release of growth factors, chemokines, cytokines and extracellular vesicles (EVs) that shuttle mRNA or miRNA to stimulate host cells’ regeneration, decrease inflammation and retard fibrosis [177,178,179,180].

Small clinical trials have reported promising proof-of-concept results regarding the safety and clinical feasibility of MSC-based therapies in AKI [181,182], diabetic nephropathy [183], atherosclerotic renovascular disease [184], lupus nephritis [185] and kidney transplant recipients [186,187,188]. MSC-based therapies need to overcome several issues, including a consistent source of cells with a stable phenotype, the optimal timing, dose and routes of administration and risks such as microvasculature collapse, rejection or tumorigenicity [189]. In this regard, MSC-derived EVs represent an alternative to MSC-based therapies that overcome most of the problems associated with MSC transplantation but retain the beneficial effects [190].

Transplantation of embryonic [191] and MSC-derived [189] EVs was beneficial in preclinical AKI and CKD [189]. MSC-derived EVs accumulated in injury sites [192], where they induced proximal tubular cell proliferation and dedifferentiation and angiogenesis. Moreover, EVs prevented renal cell apoptosis and necrosis and inhibited oxidative stress, inflammation, EMT and fibrosis [189]. The regenerative and renoprotective effects of MSC-derived EVs are meditated by their cargo in proteins, mRNAs and microRNAs. Therefore, the identification and characterization of key factors, together with the selection of best cell types for each pathological situation and the definition of the optimal protocols for EV production and isolation before clinical translation is mandatory. Nevertheless, the first clinical trial employing umbilical cord–blood–MSC-derived EVs in CKD category G3–G4 patients safely improved kidney function and decreased inflammation [193].

### 3.3. Macrophages in Repair

Macrophages present a great phenotype plasticity modulated by specific factors such as the cytokine environment. The classical macrophage phenotypes are M1 and M2 macrophages, characterized by pro- or anti-inflammatory functions, respectively [194,195]. Several beneficial functions have been assigned to M2 macrophages, such as resolution of inflammation and tissue remodeling [67,196]. However, three M2 sub-phenotypes are recognized: M2a promotes tissue repair in response to IL-4 and/or IL-13; M2b triggers host defense; and M2c is anti-inflammatory [10,197].

M2-phenotype-like macrophages participate in the later time points of kidney injury, during active tubular cell proliferation and kidney repair [12,198]. They contribute to epithelial regeneration after kidney IRI, the clearance of intraluminal debris and the attenuation of kidney inflammation [199,200,201]. Interestingly, Klotho induced M2 macrophage polarization and suppressed kidney inflammation caused by indoxyl sulfate-M1 macrophage activation in vitro [202], and the blockade of M2 polarization promoted the AKI-to-CKD transition [203]. Loss of Klotho is an early event in AKI and CKD that may be driven by albuminuria and NF-kB activation [74,204,205]. The inability of macrophages to switch between the M1 and M2 phenotypes could underlie the sustained inflammation and fibrosis developing after IRI [8]. Thus, the macrophage phenotype plays a pivotal role during kidney repair and the therapeutic use of M2 macrophages should ensure that the M2 phenotype is stable and difficult to switch in vivo. In this regard, lipocalin-2 (Lcn-2) is a potent modulator of macrophage polarization that stabilizes the M2 macrophage phenotype [206,207].

Hepatocyte growth factor (HGF) gene therapy increased the number of bone marrow-derived M2 macrophages and induced kidney repair in diabetic kidneys [208]. In a further step, cell therapy with M2 macrophages that had been stabilized with Lcn-2 effectively reduced kidney fibrosis in murine kidney disease, probably by the modulation of the renal inflammatory milieu [209,210]. Lcn-2 derived from macrophages was a major player in rescuing epithelial CCA, suggesting that Lcn-2 is a key pro-proliferative factor to rebuild damaged epithelia following injury [211].

Hence, different phenotypes of macrophages exert diverse effects in kidney disease, offering the opportunity to therapeutically regulate macrophage activation, polarization and phenotype [212,213].

### 3.4. Senescent Cell Clearance: Senotherapies

As the expansion of senescent cells promotes CKD progression, therapeutic approaches have aimed at suppressing senescent cells or their consequences. Such senotherapies include the removal of senescent cells with senolytics, SASP suppression with senostatics, or boosting blockers of cellular senescence triggers [214]. This section is focused on senolytics.

Senescent cells are resistant to apoptosis, as they undergo a switch from anti-survival to pro-survival signals, in which the B-cell lymphoma-2 (BCL-2) family of proteins plays a key role [57]. The BCL-2 family of proteins has prosurvival (e.g., BCL-2 and BCL-XL) and proapoptotic members (e.g., BAX) [33,215,216]. The BH3 domain is a key component of proapoptotic BCL-2 members and BH3-mimetics, such as ABT-737, are under study to treat malignancy by promoting cell death [217]. The BH3-mimetic Navitoclax (ABT-263) promotes the depletion of senescent human and murine renal epithelial cells [218,219]. In irradiated tubular cells, Navitoclax induced senescent cell apoptosis in a dose-dependent manner. In unilateral murine kidney IRI, Navitoclax decreased kidney injury and p21 expression and increased proliferation [219]. However, Navitoclax is undergoing clinical trials for cancer (NCT03366103) but not for CKD.

Flavonoids have been explored to deplete senescent cells in kidney injury. The combination of Dasatinib, a tyrosine kinase inhibitor which promotes apoptosis, and the pro-apoptotic natural flavonoid Quercetin has been tested in vitro and in vivo [220,221]. In murine kidney IRI, Dasatinib + Quercetin increased apoptotic cells but decreased senescent cells’ markers and collagen accumulation, thus decreasing kidney injury [221]. In a pilot study in nine patients with diabetic kidney disease, Dasatinib + Quercetin for 3 days (a “hit-and-run” therapeutic approach) reduced adipose tissue senescent cells and macrophages at 11 days [222]. A larger (*n* = 30 patients with CKD) clinical trial is exploring the impact of a 3-day course on the proportion of senescent cells in skin, fat, and/or blood (NCT02848131). These early results are encouraging, but senescent kidney cells were not assessed and the concept of the depletion of senescent kidney cells by senolytics remains untested in humans.

## 4. AKI/CKD-to-Cancer Transition

CKD patients have long been known to be at increased risk of kidney cancer. However, only recently has research focused on the contribution of AKI as an initial trigger or an aggravating factor in the chain of molecular and cellular events leading from kidney injury to kidney cancer. Thus, triggers of DNA damage may promote the expansion of pre-malignant and malignant cell clones. In this regard, both benign kidney tumors and kidney cancer may originate from renal progenitors and different types of kidney tumors can be traced to renal progenitors at specific sites of injury [223].

## 5. Conclusions

In conclusion, a better understanding of the drivers of the AKI-to-CKD transition and of the contribution to kidney repair of resident renal progenitor cells, stem cells and their reparative secretome, M2 macrophage subphenotypes within the M2 phenotype and senescent cell clearance may help identify novel biomarkers that guide therapy initiation and response to therapy as well as novel therapeutic approaches that not only prevent the AKI-to-CKD transition but also promote kidney regeneration following AKI or CKD.

## Figures and Tables

**Figure 1 ijms-23-01542-f001:**
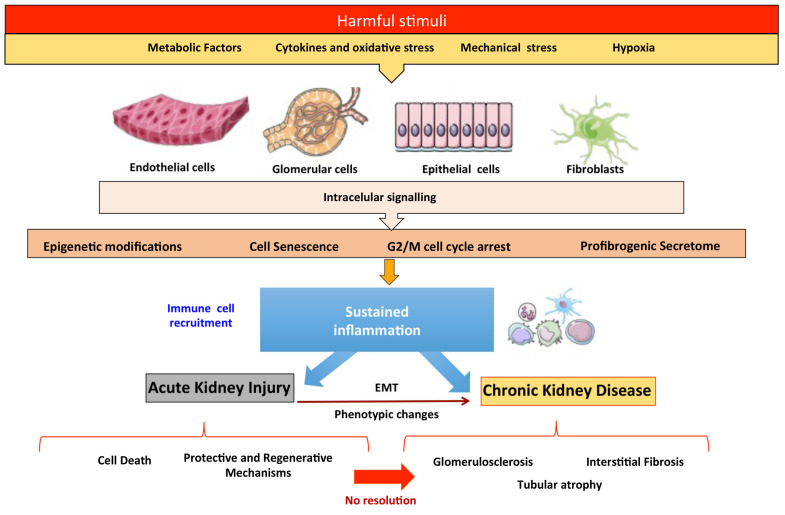
Mechanisms of kidney injury and repair.

**Figure 2 ijms-23-01542-f002:**
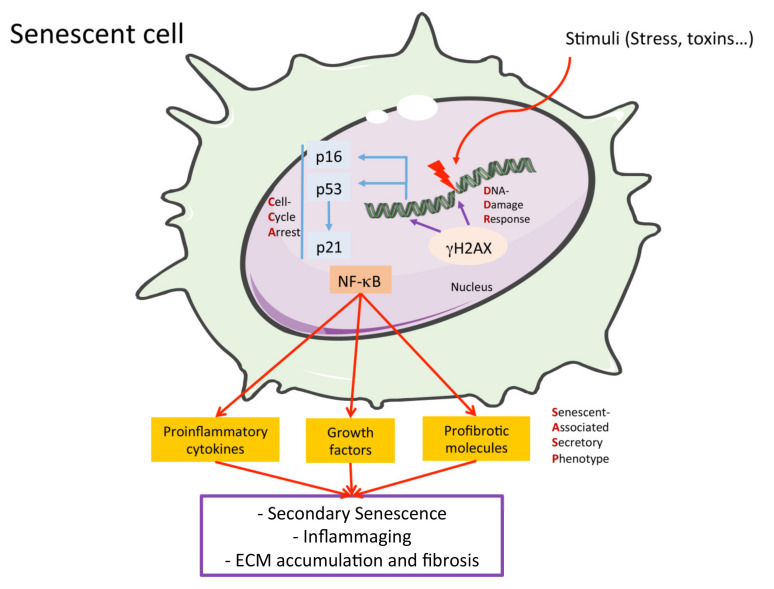
Principal hallmarks of senescent cells. Many different stimuli may cause a DNA damage in the cell and activate the DNA-Damage Response (DDR), which may produce the expression of the Cell-Cycle Arrest (CCA) molecules. Prolonged activation of DDR and CCA proteins produce cellular senescence and the release of the Senescent-Associated Secretory Phenotype (SASP), enriched in different proinflammatory, profibrotic and growth factors that are regulated by NF-κB pathway activation and may cause activation of cellular senescence in a paracrine manner, inflammaging and fibrosis in the kidney, and other tissues.

**Figure 3 ijms-23-01542-f003:**
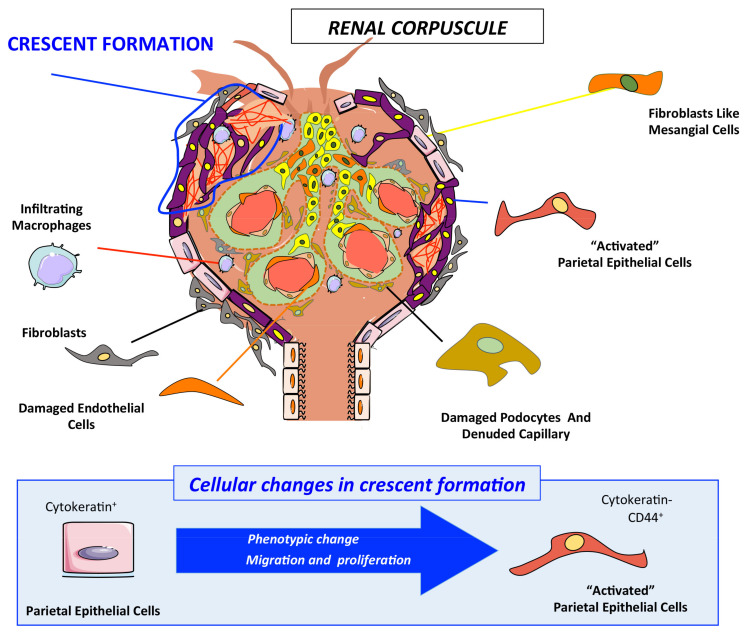
Glomerulonephritis characterized by a fibrous structure formation (crescent) in renal corpuscle. Glomerular PECs (cytokeratin+) are the main actors, infiltrating cells such as macrophages (CD68+) in crescent formation during rapid progressive glomerulonephritis/crescentic glomerulonephritis (RPGN/CGN). Activated PECs (CD44+/cytokeratin-) invade the glomerular urinary space and obstruct the urinary space. PEC “activation” implies a phenotypic change that promotes their abnormal proliferation and their migration towards the urinary space of renal corpuscles.

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
