# Peer review of "Molecular Mechanisms of Kidney Injury and Repair"

_ijms, 2022, doi:10.3390/ijms23031542_

Round 1

Reviewer 1 Report

The manuscript is a high-quality and well-written review on molecular mechanisms of kidney injury and repair. 

I have only one suggestion: 

Numerous epidemiologic studies report the association between CKD and kidney cancer. Furthermore, recent studies suggest that kidney cancer develops following an AKI episode or after years of CKD at the stage of kidney failure as a possible maladaptive repair mechanism to injury (Kidney Int. 2021;100(1):55-66). For that reason, I suggest adding a paragraph on molecular and cellular events contributing to AKI/CKD-to-Cancer transition. 

The quality of the figures needs to be improved.

Author Response

ANSWER TO REFEREES

We thank the reviewers for their critical comments to our manuscript entitled “Molecular Mechanisms of Kidney Injury and Repair” that helped to improve the quality of both content and presentation of our work.

Please find here the point-by-point response to the referee comments. We have introduced the corresponding modifications in the text and we generated a new version of the manuscript.

REVIEWER 1.-

The manuscript is a high-quality and well-written review on molecular mechanisms of kidney injury and repair.

I have only one suggestion:

1.1.- Numerous epidemiologic studies report the association between CKD and kidney cancer. Furthermore, recent studies suggest that kidney cancer develops following an AKI episode or after years of CKD at the stage of kidney failure as a possible maladaptive repair mechanism to injury (Kidney Int. 2021;100(1):55-66). For that reason, I suggest adding a paragraph on molecular and cellular events contributing to AKI/CKD-to-Cancer transition.

Response: We agree that this is a very timely topic, but a detailed review is beyond the scope of the present manuscript. As the reviewer points out, an excellent recent review has summarized knowledge on the topic. We have added a brief section to the manuscript (page 11)  to acknowledge its existence and listed potential mechanisms involved, without delving into further detail. We hope this is what the reviewer was expecting.

1.2.- The quality of the figures needs to be improved.

Response: As suggested, we have improved the final quality of the figures.

Reviewer 2 Report

The paper "Molecular Mechanisms of Kidney Injury and Repair" is properly written  following typical structure for scientific review. 

Introduction on kidney injury is very short. From practical point of view pre-renal (mainly ischemia), renal (GN, neprotoxicity) and post-renal mechanism of injury should be considered. They differ in course and risk of transition into CKD. For the instance decopression of hydronephrosis in solitary kidney result in polyuria which usualy is not risky for CKD if scuccessfully terated.
Could you address different way of kidney injury?

The second part "Molecular and cellular events contributing to the AKI-to-CKD transition" is much more developed. 

Some renal patients show "progressor" phenotype after experiencing AKI. Can we nowadays identify them using molecular markers? Do we need repeated biopsy?

Author Response

REVIEWER 2.-

The paper "Molecular Mechanisms of Kidney Injury and Repair" is properly written  following typical structure for scientific review. 

2.1.- Introduction on kidney injury is very short. From practical point of view pre-renal (mainly ischemia), renal (GN, neprotoxicity) and post-renal mechanism of injury should be considered. They differ in course and risk of transition into CKD. For the instance decopression of hydronephrosis in solitary kidney result in polyuria which usualy is not risky for CKD if scuccessfully terated.   
Could you address different way of kidney injury?   

The second part "Molecular and cellular events contributing to the AKI-to-CKD transition" is much more developed. 

Response: As suggested, we have expanded on the causes and consequences of AKI and CKD.

2.2.- Some renal patients show "progressor" phenotype after experiencing AKI. Can we nowadays identify them using molecular markers? Do we need repeated biopsy?

Response: Now we have add a sentence stating the problem  in the introduction.

Round 2

Reviewer 1 Report

I have no further recommendations for the authors.